# Transcriptomic Changes Associated with *ERBB2* Overexpression in Colorectal Cancer Implicate a Potential Role of the Wnt Signaling Pathway in Tumorigenesis

**DOI:** 10.3390/cancers15010130

**Published:** 2022-12-26

**Authors:** Eman A. Abdul Razzaq, Khuloud Bajbouj, Amal Bouzid, Noura Alkhayyal, Rifat Hamoudi, Riyad Bendardaf

**Affiliations:** 1Sharjah Institute for Medical Research, University of Sharjah, Sharjah P.O. Box 27272, United Arab Emirates; 2Department of Clinical Sciences, College of Medicine, University of Sharjah, Sharjah P.O. Box 27272, United Arab Emirates; 3Oncology Unit, University Hospital Sharjah, Sharjah P.O. Box 72772, United Arab Emirates; 4Division of Surgery and Interventional Science, University College London, London WC1E 6BT, UK

**Keywords:** *ERBB2*, HER2, colorectal cancer, RNA-seq, whole transcriptomic analysis, NKX2-5, Wnt signaling

## Abstract

**Simple Summary:**

The present study identified cellular pathways and genes co-expressed with HER2 in colorectal cancer using whole transcriptomic analysis on colorectal cancer patients and cell lines. A comparison of the genes and pathways between patients and cell lines identified the Wnt signaling pathway and the homeobox gene NKX2-5 to be significant. This study sheds new light on the role of HER2 in colorectal cancer pathogenesis.

**Abstract:**

Colorectal cancer (CRC) remains the third most common cause of cancer mortality worldwide. Precision medicine using OMICs guided by transcriptomic profiling has improved disease diagnosis and prognosis by identifying many CRC targets. One such target that has been actively pursued is an erbb2 receptor tyrosine kinase 2 (*ERBB2*) (Human Epidermal Growth Factor Receptor 2 (HER2)), which is overexpressed in around 3–5% of patients with CRC worldwide. Despite targeted therapies against HER2 showing significant improvement in disease outcomes in multiple clinical trials, to date, no HER2-based treatment has been clinically approved for CRC. In this study we performed whole transcriptome ribonucleic acid (RNA) sequencing on 11 HER2+ and 3 HER2− CRC patients with advanced stages II, III and IV of the disease. In addition, transcriptomic profiling was carried out on CRC cell lines (HCT116 and HT29) and normal colon cell lines (CCD841 and CCD33), ectopically overexpressing *ERBB2*. Our analysis revealed transcriptomic changes involving many genes in both CRC cell lines overexpressing *ERBB2* and in HER2+ patients, compared to normal colon cell lines and HER2− patients, respectively. Gene Set Enrichment Analysis indicated a role for HER2 in regulating CRC pathogenesis, with Wnt/β-catenin signaling being mediated via a HER2-dependent regulatory pathway impacting expression of the homeobox gene NK2 homeobox 5 (NKX2-5). Results from this study thus identified putative targets that are co-expressed with HER2 in CRC warranting further investigation into their role in CRC pathogenesis.

## 1. Introduction

Despite improvements in early detection and treatment methods during the last two decades, colorectal cancer (CRC) remains the 3rd most common cause of cancer-related death worldwide with approximately 150,000 new CRC cases diagnosed in the United States annually [1]. Among them, approximately 20% of patients will have distant metastasis, and around 30% of patients with stage II and III disease will develop metastasis [2]. The five-year survival rate of CRC patients with distant metastasis is less than 15% [2]. The incidence of CRC in men and women under the age of 50 has steadily increased in the past two decades [1,2]. For patients with metastatic CRC (mCRC), chemotherapy remains the mainstay of treatment, but eventually, all patients develop resistance to therapy and experience treatment failure due to the intra-tumoral heterogeneity of CRC [3].

Precision medicine guided by tumor genomic profiling has transformed the cancer diagnosis, prognosis, and treatment paradigm over the past two decades. However, recent estimates suggest that fewer than 10% of cancer patients benefit from this approach [4,5]. The primary issues facing genomic profiling include firstly, actionable genomic alterations are not detected in a vast majority of cases [6], and even when they are, secondly, a significant proportion of patients fail to experience an antitumor response to the indicated targeted therapy [7].

One such target is the amplification of *ERBB2* (HER2), which occurs in approximately 3% of patients with metastatic CRC (mCRC) and 5% of patients with wild-type *NRAS* and *KRAS* tumors [8,9]. Several *ERBB2*-targeted therapies are either in different phases of clinical trials or approved for use in patients with *ERBB2*-positive breast and gastric and gastroesophageal tumors [8]. However, despite recommendations of the National Comprehensive Cancer Network guidelines and clinical evidence from phase II trials that anti-*ERBB2* therapies improve disease outcomes in *ERBB2*-positive mCRC patients, no *ERBB2*-directed approved therapies for patients with CRC are currently approved for clinical use [8,9,10].

The role of HER2 in carcinogenesis is most well-characterized in breast cancer [11,12]. HER2+ breast cancer is a historically aggressive subtype of breast cancer with a five-year survival rate of 30% [11,13,14]. The discovery that amplification or overexpression of *ERBB2* was associated with extremely poor survival in breast cancer led to efforts that resulted in the development of a monoclonal antibody (mAb) to HER2, trastuzumab [14,15]. However, whether *ERBB2* overexpression-mediated carcinogenesis follows similar mechanisms in breast and colon tissue is unknown.

Thus far, the majority of the studies related to *ERBB2* in cancers have focused on identifying the landscape of genomic amplification in *ERBB2* and defining therapeutic regimens to target these amplifications [8]. Our earlier study has shown that *ERBB2* mRNA and protein overexpression correlates with more aggressive colorectal cancer in the North African population [16]. However, the effect of the overexpression of *ERBB2* on the global transcriptomic profiles within CRC patients is not known. Therefore, the objective of the current study is to characterize whole transcriptomic changes associated with *ERBB2* overexpression in CRC cell lines and patient samples with a view to gain a deeper understanding of the role of *ERBB2* in CRC pathogenesis.

## 2. Materials and Methods

### 2.1. Patients and Tissue Specimens

Ethical approval for the study was provided by the Research and Ethics Committee (REC) of University Hospital Sharjah (UHS-HERC-055-25022019). All methods were performed in accordance with the relevant guidelines based on the Declaration of Helsinki and the Belmont Report. We obtained written informed consent from all study participants. This is a retrospective study of 14 patients with primary CRC. Patients with secondary cancers were excluded, whilst all primary CRC patients were included, regardless of age, gender, or tumor stage. The initial diagnosis was performed prior to and independently of our study to determine the Tumor, lymph Nodes, and Metastasis (TNM) score. Tissues were sectioned from formalin-fixed, paraffin-embedded (FFPE) biopsies for molecular and immunohistochemical analysis.

### 2.2. Immunohistochemistry

To begin with, 3 μm sections from the FFPE of 14 CRC patients’ biopsies were immunohistochemically stained using the rabbit monoclonal antibody for HER2 (1:4000 dilution; ab214275, Abcam, Waltham, MA, USA) according to the manufacturer’s instructions. An experienced pathologist (R.H.) scored the stained slides, following the consensus recommendations for HER2 scoring for CRC [17,18]. Briefly, scoring was performed on a 4-point scale—0, 1+, 2+, 3+ focusing on intensity and extent according to the Allred scoring system [19]. In this study, 0 and 1+ intensity were taken to be negative for HER2 expression and the study focused on the assessment of membranous HER2 expression.

### 2.3. Cell Culture

The CRC cell lines HT29 and HCT116, and two normal colon cell lines: CCD33 and CCD841, were obtained from Bio Medical Scientific Services (BIOMSS, Al Ain, United Arab Emirates), and cultured in Dulbecco’s Modified Eagle’s Medium (DMEM) supplemented with 10% fetal bovine serum (Sigma Aldrich, St. Louis, MO, USA), 1% Penicillin/Streptomycin (Sigma) and 20 mM L-Glutamine (Sigma) at 37 °C in 5% CO_2_ incubator.

### 2.4. Transfection

The wild-type *ERBB2* expression construct was a gift from Mien-Chie Hung (Addgene plasmid #16257; https://www.addgene.org/16257, accessed on 15 December 2020) [20]. The cell lines were transfected with 5µg of pcDNA3-*ERBB2* plasmid construct using Lipofectamine 3000 reagent (ThermoFisher Scientific, Cambridge, MA, USA) according to the manufacturer’s instructions. Cells transfected with the empty pcDNA3 vector served as the experimental control. The *ERBB2* expression level was checked 24 h post-transfection at the mRNA and protein levels using qRT-PCR and Western blotting, respectively.

### 2.5. RNA Isolation

RNA extraction was carried out from three sequential (3 µm) sections from the same FFPE block. A needle macrodissection was carried out to enrich the tumor’s content. This was carried out by marking the tumor areas on the slides and carefully removing the unmarked non-tumor areas using a sterile needle, following which the marked areas were collected for molecular analysis. RNA was extracted using the RNA RecoverAll kit (ThermoFisher Scientific, Cambridge, MA, USA) according to the manufacturer’s instructions. Genomic DNA removal was ensured by treating the RNA with Turbo DNase (ThermoFisher Scientific). For RNA extraction from cell lines, cells were pelleted at 12 × 103 g for 5 min and rinsed thrice with ice-cold 1X sterile PBS. RNA was extracted from the cell pellet as described above using the RNA RecoverAll kit, followed by genomic DNA removal using Turbo DNase. All RNA samples were stored at −800 °C until further use.

### 2.6. Quantitative Reverse Transcriptase-PCR (qRT-PCR)

qRT-PCR was performed using the Superscript First-strand Synthesis system (ThermoFisher Scientific). Real-time qPCR was performed in triplicates, using SYBR green (Solis BioDyne, Tartu, Estonia), on Quant Studio 3 (Applied Biosystem, Waltham, MA, USA). *ERBB2* and the reference genes (18S ribosomal RNA) were pre-amplified using the following primer sets [18]. *ERBB2*_sense: 5′-ACATGCTCCGCCACCTCTACCA-3′; *ERBB2* Antisense: 5′-GGATCTGCCTCACTTGGTTGTG-3′; 18SrRNA_sense: 5′-TGACTCAACACGGGAAACC-3′; 18SrRNA_antisense: 5′-TCGCTCCACCAACTAAGAAC-3′. A total of 40 PCR cycles were performed consisting of 15 s denaturation at 95 °C and a combined annealing and extension cycle of 10 min at 60 °C. The threshold cycle value (Ct) was normalized against the Ct value of internal control 18 s RNA.

### 2.7. Cell Lysis and Western Blot

Cell lysis and Western blot were performed as described previously [21]. Anti-HER2 (1:1000; Abcam) was used to probe the blots. All blots were subsequently stripped and re-probed for β-Actin (1:5000; Abcam, Waltham, MA, USA) to confirm equal loading.

### 2.8. Next-Generation RNA Sequencing

RNA sequencing was carried out on the indicated samples using a targeted AmpliSeq Transcriptome panel on Ion S5 XL System (ThermoFisher Scientific, Cambridge, MA, USA). In brief, ~30 ng of Turbo DNase treated RNA was used for cDNA synthesis using a SuperScript VILO cDNA Synthesis kit (ThermoFisher Scientific) followed by amplification using Ion AmpliSeq gene expression core panel primers. The enzymatic shearing was performed using FuPa reagent to obtain amplicons of ~200 bp and the sheared amplicons were ligated with the adapter and the unique barcodes. The prepared library was purified using Agencourt AMPure XP Beads (Beckman Coulter, Indianapolis, IN, USA) and the purified library was quantified using an Ion Library TaqMan™ Quantitation Kit (Applied Biosystems, Waltham, MA, USA). The libraries were further diluted to 100 pM and pooled equally with four individual samples per pool. The pooled libraries were amplified using emulsion PCR on Ion OneTouch™ 2 instrument (OT2) and the enrichment was performed on Ion OneTouch™ ES following the manufacturer’s instructions. Thus, prepared template libraries were then sequenced with Ion S5 XL Semiconductor sequencer (ThermoFisher Scientific, Cambridge, MA, USA) using the Ion 540™ Chip.

### 2.9. Bioinformatics Analyses

RNA-seq data were analyzed using Ion Torrent Software Suite version 5.4 and the alignment was carried out using the Torrent Mapping Alignment Program (TMAP). TMAP is optimized for aligning the raw sequencing reads against the reference sequence derived from the hg19 (GRCh37) assembly, and the specificity and sensitivity were maintained by implementing a two-stage mapping approach by employing BWA-short, BWA-long, SSAHA [22], Super-maximal Exact Matching [23] and the Smith–Waterman algorithm [24] for optimal mapping. Raw read counts of the targeted genes were performed using Samtools (Samtools view–c–F 4–L bed_file bam_file) and the number of expressed transcripts was confirmed after Fragments Per Kilobase Million (FPKM) normalization. For technical variations, code-set content normalization was performed with the geometric median for all genes. Principal component analysis (PCA) was performed using the indicated samples with R statistical software. Differentially expressed gene (DEG) analysis was performed using R/Bioconductor package DESeq2 with raw read counts from the RNA sequencing data [25,26]. Genes with less than ten normalized read counts were excluded from further analysis. A fold change of 2 was set as the cutoff for differentially expressed gene identification. *p* < 0.05 was considered statistically significant. The DEGs were then subjected to Gene Set Enrichment Analysis (GSEA).

### 2.10. Analyses of Publicly Available Transcriptomic Data Sets for Breast Cancer

In order to compare the biological pathways and differentially expressed genes between the *ERBB2* over-represented in breast cancer (BC) and CRC, transcriptomic data sets of BC were searched and retrieved from the GEO (Gene Expression Omnibus) database (https://www.ncbi.nlm.nih.gov/geo/, accessed on 3 October 2022). The datasets were searched based on “Breast cancer” and “*ERBB2*” keywords. Then, datasets including BC patients with variable *ERBB2* expression based on the same platform Affymetrix Human Genome U133 Plus 2.0 Array were considered. Sixteen well-matched datasets were available, out of which fourteen were excluded for further analysis. Exclusion criteria were datasets performed *in vitro* cancer cell lines or in vivo study models using non-human species, repeated samples in super-series, and datasets exhibiting poor *ERBB2* expression values uncharacteristic of HER2+/HER2−. Two datasets that met the criteria were selected, including GSE29431 and GSE48391 (Appendix A). The raw data and the probe annotation files were downloaded for further analysis.

### 2.11. Breast Cancer Microarray Data Analysis

A total of 65 BC patients were selected in our analysis including 48 samples with low *ERBB2* and 17 samples with high *ERBB2* expression. The Affymetrix microarray represents more than 54,000 probes where each gene is represented with different probes. The raw data were processed and normalized using in-house R script as previously described [27]. For normalization and adaptive filtering, Affymetrix Microarray Suite 5 (MAS5) and Gene Chip Robust Multiarray Averaging (GCRMA) packages in Bioconductor/R software were applied. Probes with a MAS5 value > 50 and coefficient of variation (CV) 10–100% in GCRMA among all samples of each dataset were identified to get only common variant probes. The filtered probes were then annotated and collapsed into the gene names list based on the maximum expression of probes for each gene. The unchanged probes, positive control probes, and unassigned probes were excluded from the downstream analysis. The mapped gene expression lists were subjected to Gene Set Enrichment Analysis (GSEA) to identify the activated and enriched biological pathways between high and low *ERBB2*-BC patients.

### 2.12. GSEA

GSEA was carried out separately for all resulting gene sets from the above different transcriptomic analyses including CRC patients, CRC cell lines, normal colon cell lines, and BC patients. First, the absolute GSEA was performed to identify the significantly enriched pathways among sets related to the C2: curated gene sets; C5: ontology gene sets including molecular function (MF) and biological process (BP); C6: oncogenic signature gene sets; and C7: immunologic signature gene sets. The results of the GSEA were ranked and selected according to the *p* < 0.05 as described previously [27,28]. Next, the selected significant pathways were further analyzed to identify the differentially enriched genes and the leading edge genes in each pathway. In order to further reduce the set of resulting genes, a systematic cross-reference of each gene enriched within statistically significant pathways was carried out. Finally, the genes with the highest frequency across the multiple significant pathways enriched between the HER2 positive and HER2 negative samples were identified.

### 2.13. Statistical Analysis

Functional data are presented as mean ± SD, except where otherwise stated. When two groups were compared, the student’s *t*-test was used unless otherwise indicated. *p* < 0.05 was considered as statistically significant.

## 3. Results

### 3.1. Overexpression of ERBB2 Induces Distinct Transcriptional Profiles in the CRC Cell Lines HT29 and HCT116

We wanted to determine if ectopic overexpression of *ERBB2* in CRC cell lines induces genome-wide transcriptional changes. Hence, we screened different CRC cell lines to identify cell lines with low endogenous *ERBB2* expression. We initially determined the steady-state expression of *ERBB2* mRNA in the normal colon cell lines, CCD33 and CCD841, as well as the CRC adenoma cell lines HCT116 and HT29 using qRT-PCR. Compared to CCD33 and CCD841 cells, *ERBB2* expression was 6.28 ± 0.003 folds and 7.56 ± 1.54 folds less in HCT116 and 3.71 ± 0.01 4.43 ± 0.02 folds less in HT29, respectively (Figure 1A).

The CRC cell lines HT29 and HCT116 and normal colon cell lines CCD33 and CCD841 were transiently transfected with either an empty pcDNA3 vector or *ERBB2* expression construct. Successful transfection was confirmed by both qRT-PCR (Figure 1B), and Western blot analyses (Figure 1C).

RNA isolated from HCT116, HT29, CCD33, and CCD841 cells expressing an empty pcDNA3 vector (control) and those expressing ectopic *ERBB2* (*ERBB2*) were then subjected to RNA-seq in biological replicas. Multidimensional scaling using PCA was performed. Clusters distinguished by *ERBB2* expression levels in HCT116 and CCD841 cells were observed, confirming the reproducibility of the replicates and the unique transcriptomic profile associated with the ectopic expression of *ERBB2*; whereas samples were more staggered for the HT29 and CCD33 cells (Appendix A). The volcano plot of these data exhibited robust *ERBB2* whole transcriptomic changes in the CRC cell lines; HCT116 (1774 DEGs—730 upregulated and 1044 downregulated) and HT29 (1289 DEGs—430 upregulated and 859 downregulated) compared to the normal colon cell lines CCD33 (160 DEGs) and CCD841 (312 DEGs), (Appendix A). In addition, the unsupervised hierarchical clustering analysis using the DEGs from each comparison exhibited clear subgroups of *ERBB2* transfected and control cell lines, for HCT116, HT29, CCD33, and CCD841 (Figure 2 and Appendix A). The DEGs lists resulting from comparing *ERBB2* overexpression and control samples of HCT116, HT29, CCD33, and CCD841 are listed in Appendix A.

### 3.2. Global Transcriptional Profiling in CRC Patients Based on HER2 Differential Expression

Given that cell lines exhibit a homogenous system, and a 2-D culture might not be a true replicate of an actual tumor, we next determined the genome-wide transcriptional patterns in CRC patients with varying HER2 protein expression. Of the 14 patients that were recruited, there were 8 females and 4 males with ages ranging from 37 to 86 years (mean ± SD, 63.15 ± 15.02 years). Histopathological examination identified most cases as adenocarcinomas.

Following the HER2 diagnostic criteria, 0 and 1+ staining scores were considered negative [29]. Among the CRC cases tested, 78.57% (11/14) of the cases were positive for HER2 (Score ≥ 2+; Figure 3A) whereas 21.43% (3/14) were negative for HER2 (≤1+; Figure 3B). All clinicopathological data are shown in Appendix A. No difference in classification and staging was observed with respect to gender, age, or HER2 expression.

RNA isolated from the 3 HER2− and 11 HER2+ biopsies were then subjected to whole RNA sequencing. Multidimensional scaling using PCA revealed clusters distinguished by *ERBB2* expression levels, confirming the unique transcriptomic profile associated with differential *ERBB2* expression (Figure 4A). The volcano plot of these data identified groups of differentially expressed genes, showing that 2701 were differentially expressed in HER2+ compared to HER2− CRC-patients of which 1344 were upregulated and 1357 were downregulated (Figure 4B). Additionally, the unsupervised hierarchical clustering analysis based on the total DEGs showed that HER2+ and HER2− CRC-patients are, respectively, clustered as a single branch, further confirming the distinct transcriptional profiling between HER2+ and HER2− CRC-patients (Figure 4C, Appendix A). The differential HER2 protein expression in the 14 patient biopsies was further confirmed by the relative *ERBB2* gene expression in these samples (Appendix A). The DEGs between HER2+ and HER2− CRC patients are listed in Appendix A.

### 3.3. GSEA of DEGs Revealed Distinctive ERBB2-Mediated Activation of Various Cellular Pathways Including Wnt Signaling and Regulation of Cellular Differentiation

To perform a functional interpretation of our transcriptomic analyses, the DEGs genes resulting from each comparison between HER2+ and HER2− CRC patients, HER2+ and control CRC cell lines, and HER2+ and normal colon cell lines were initially used as the input for the GSEA to identify the significantly enriched pathways (*p* < 0.05) among gene sets related to the following: C2, C5 (BP and MF), C6, and C7 collections. Given that the aim of the study was to define the putative role of HER2 in CRC pathogenesis, we chose the gene sets that contain functional pathways linked to cancer hallmarks and immune response. The results identified 98 significantly differentially activated pathways in HER2− compared to HER2− CRC patients (Appendix A). The most significant pathways included the Wnt signaling pathway, T cell receptor signaling pathway, cell cycle, and cell differentiation pathways. Likewise, in HCT116 and HT29 CRC cell lines, the GSEA results showed, respectively, about 15 and 89 significant molecular functions and biological processes ontology gene sets (Appendix A). Pathways related to cell signaling and leukocyte differentiation and migration were enriched in the HER2+ CRC cell lines. However, as expected, only a few significant activated cellular pathways were enriched in HER2+ compared to control normal colon cell lines (Appendix A). Moreover, by overlapping the enriched pathways in CRC patients and cell lines, the results showed that the regulation of ion transport, cell–cell signaling, and cell proliferation were overexpressed between the two CRC systems.

An analysis of the leading-edge genes of the significant gene sets within the patients revealed that many were consistently represented by counting the number of times a gene occurs (gene frequency) across all the different pathways, suggesting that these genes strongly influenced the HER2-mediated expression pattern. The top 20 genes based on the gene frequency in the significantly activated cellular pathways between HER2+ and HER2− CRC patients showed key genes, including *NKX2-5, NKX6-1, WNT3A, WNT5A, NOG, SOX9, SOX18* (Figure 5A). The functional annotation of the top leading-edge genes showed highly significant enrichment of categories related to the regulation of cell differentiation, canonical Wnt signaling, cell development and maturation, and regulation of epithelial cell differentiation (Figure 5B).

### 3.4. Comparison of Cellular Pathways Revealed ERBB2-Mediated Enrichment of Pathways Related to Stem Cell Differentiation, Regulation of Wnt Signaling, and Immune Activation in Both Colorectal and Breast Cancers

Given the well-characterized role of *ERBB2* in breast cancer pathogenesis, we next analyzed breast cancer datasets available within the GEO database to determine if there are any similarities in the *ERBB2*-mediated enrichment of pathways in breast and colon cancer patients. Two independent datasets from different populations were selected: GSE29431 Caucasian/Spanish and GSE48391 Asian/Chinese (Appendix A). These two datasets did not include the IHC scores for HER2. When comparing our transcriptomics data with the HER2 IHC, we found them to correlate if we take the average transcriptomics expression of the *ERBB2* mRNA. Therefore, we attempted to perform the stratification of HER2 from publicly available resources in a similar manner to the way we did with our patients’ cohort from the mRNA transcriptome data, by taking the average HER2 expression from each group and stratifying them as HER2+ and HER2− accordingly. Therefore, followed by a quality control assessment based on HER2 expression in a particular patient compared to average HER2 expression in all patients within each cohort, 13 HER2− and 8 HER2+ were selected from the GSE29431 dataset and 35 HER2− and 9 HER2+ were selected from the GSE48391 dataset (Appendix A).

The GSEA was carried out to identify the significantly enriched pathways between HER2+ and HER2− breast cancer (BC) patients amongst gene sets related to the following: C2, C5 BP and MF, C6, and C7. Different significantly activated pathways and ontology gene sets were identified between HER2+ and HER2− BC patients (Appendix A). A comprehensive comparison between the significantly enriched pathways in HER2+/− BC and CRC patients was performed. The most significantly enriched pathways that are unique to BC patients include the VEGF signaling pathway, regulation of kinase activity MAPK pathway, and regulation of steroid metabolic process. On the other hand, twenty-six common activated pathways were observed between HER2+/HER2− breast cancer and CRC patients (Appendix A). This included pathways related to stem cell differentiation, regulation of Wnt signaling, and 17 related to immunological signature subsets including predominantly T cells, macrophages, and NK activation as depicted in Figure 6. In addition, the enrichment analysis identified unique immune-related pathways including macrophage activation and T-cell response, as shown in Appendix A. Analysis of the leading-edge genes underlying the enrichment of each gene set within the BC and CRC patient datasets (Figure 7A) revealed great resemblance in both lists of top genes (7 out of the top 20 genes) involved in the 26 common activated pathways of HER2+ vs. HER2− patients, and the top 20 frequent genes in activated cellular pathways between HER2+ and HER2− CRC patients (Figure 7B). The functional annotation of the top 20 genes between both colorectal and breast cancers revealed significant enrichment of categories related to the regulation of stem cell differentiation, protein catabolic process, regulation of peptidase and hydrolase activity, and regulation of Wnt signaling (Figure 7C). Moreover, by comparing the different enriched pathways between HER2+/− CRC patients, HER2+/− BC patients and cell lines, we noticed that, importantly, the response to calcium ions is a common enriched biological pathway between CRC patients, BC patients and CRC cell lines but absent in normal colon cell lines.

## 4. Discussion

Our earlier work has shown that HER2 overexpression is correlated with more aggressive disease in CRC patients [16], indicating that stratification of patients according to HER2 status might be beneficial in the early detection and subsequent therapeutic management of patients with metastatic CRC. The results from the current study reveal that HER2 overexpression is associated with distinct global transcriptomic profiling, the characterization of which might lead to the identification of putative diagnostic and prognostic biomarkers.

In our study, there were fewer up and downregulated genes in HT29 compared to HCT116, indicating that the transcriptomics perturbation post-HER2-overexpression was less in HT29 in comparison to HCT116. The expression levels of APC, GSK3B, and CTNNB1 were not significant, which indicates that perhaps there may be a link between *ERBB2* expression and TP53 mutational status (rather than APC), since HT29 harbors TP53 mutation.

About half of the top 100 differentially expressed genes between HER2+ and HER2− CRC patients exhibited a more prominent difference in expression and were overexpressed in the HER2+ samples. A vast majority of them were non-coding RNA (small nucleolar RNAs, snoRNAs) that are normally involved in the biogenesis of other RNAs. That they were among the top differentially expressed genes indicates that expression of the snoRNAs may be regulated by HER2 and that they might be involved in the CRC pathogenesis in HER2+ patients. Indeed, it has been reported that numerous snoRNAs, including tumor-promoting and tumor-suppressing snoRNAs, are not only dysregulated in tumors but also show associations with clinical prognosis [30]. In addition, aberrant expression of snoRNAs has been reported in cell transformation, tumorigenesis, and metastasis, indicating that snoRNAs may be considered as biomarkers and/or therapeutic targets of cancer [31]. Even CRC associations between snoRNAs and CRC development have been reported [32,33,34]. For example, SNORA21 promotes CRC cell proliferation by regulating cancer-associated pathways such as Hippo and Wnt signaling pathways, and overexpression of SNORA21 has been reported to be associated with distant metastasis in CRC [35].

Colorectal cancer demonstrates hyperactivation of the Wnt pathway, which is involved in tumorigenesis, stemness, and metastatic progression [36,37]. One of the enriched pathways in our analysis was the regulation of stem cell differentiation. HER2-overexpression in gastric cancer cells results in increased stemness and invasiveness [38]. Furthermore, this increased HER2-mediated stemness is regulated by Wnt/β-catenin signaling [39]. Our analysis also revealed significant enrichment of pathways related to positive and negative regulation of Wnt signaling and the enrichment of the Wnt signaling pathway genes *WNT3A* and *WNT5A*. These results would thus indicate that HER2 overexpression in CRC cells might result in poorly differentiated tumors that are more invasive.

One of the leading-edge genes that were differentially expressed between HER2+ and HER2− patients and cell lines was a homeobox gene of the NKL subclass, cardiac transcription factor *NKX2-5*. *NKX2-5* is one of the earliest known transcription factors required for cardiac cell specification and proliferation [40,41,42].

*NKX2-5* is expressed in several types of tumors [43,44,45,46], but its precise role in tumorigenesis is unknown. Another family member of the NKL, *NKX2-1*, has been reported to mediate p53-induced tumor suppression [47,48,49,50], Indeed, in the context of CRC, *NKX2-5* functions as a conditional tumor suppressor gene via activating the p53-mediated p21WAF1/CIP1 expression [51]. It has been predicted via bioinformatic analysis and confirmed by chromatin immunoprecipitation analysis that in hepatocellular carcinoma cells the promoter of *ERBB2* binds to the transcription factor *NKX2-5*, resulting in a negative regulatory effect [52]. Interestingly, promoter hypermethylation of *NKX2-6* has been identified as a candidate biomarker associated with differential methylation in HER2+ breast cancer and breast carcinogenesis [53]. Whether a similar mechanism is operant for the HER2-mediated downregulation of *NKX2-5* remains to be determined.

During cardiogenesis, *NKX2-5* potentiates Wnt signaling by regulating the expression of the R-spondin3 [54]. In the current study, we also saw the enrichment of pathways related to positive and negative regulation of Wnt signaling. Analysis of the RNAseq data showed significantly more *NKX2-5* expression in HER2− patients compared to HER2+ patients (Appendix A). Whether HER2 overexpression drives negative regulation of *NKX2-5*-mediated Wnt signaling, ultimately resulting in well-differentiated less invasive tumors in CRC, remains to be determined.

It is important to note that the transcriptomic analysis in CRC cell lines was conducted post-transient transfection in CRC cell lines. It is highly plausible that short-term transfection (to an unphysiological expression level) in the CRC cell lines will induce transcriptomic changes that will be different from those observed in cell lines that have evolved under HER2-overexpression selective pressure. Hence, comparing our data to gene expression changes observed in HER2-amplified or HER2-mutant cell lines would indeed be informative in this context. However, this shortcoming is potentially offset to a large extent by three facts—(a) our analysis also involved HER2− and HER2+ patient samples, which are a better model compared to any of the homogeneous cell line models; (b) our usage of two different CRC cell lines and two normal colon cell lines; and (c) our observation that similar pathways were being enriched when the patient samples and the HER2-overexpressing cell lines were compared.

Despite the current study having inherent weaknesses in low sample size and lack of validation in a wider population, the results do highlight the importance of transcriptional profiling of HER2+ and HER2− CRC patients in identifying potential biomarkers that play a role in CRC pathogenesis. Additional studies are warranted in different population cohorts as the incidence of HER2+ CRC patients varies widely based on geographical location. It would be intricate and intriguing to investigate whether and how HER2 regulates *NKX2-5* in wild-type and p53 mutant CRC and their subsequent effects on Wnt signaling and CRC invasives.

## 5. Conclusions

The study identified genes and pathways related to HER2 expression in colorectal cancer in both patients and colorectal cancer cell lines. Comparison of whole transcriptome RNA sequencing analysis in patients with HER2+ and HER2− identified unique immune-related pathways including macrophage activation, and T-cell response as well as cancer-related pathways including Wnt signaling and MYC-related pathways. The Wnt signaling pathway was also significant in the two different colorectal cell lines (HCT116 and HT29) when compared with normal colon cell lines (CCD841 and CCD33), ectopically overexpressing *ERBB2*. Differential gene expression analysis identified many Wnt signaling pathway genes including *WNT5A, WNT9A, WNT3A, WNT16*, and *WNT10A*. Gene Set Enrichment Analysis indicated a role for HER2 in regulating CRC pathogenesis, with Wnt/β-catenin signaling being mediated via a HER2-dependent regulatory pathway impacting expression of the homeobox gene *NKX2-5*. Results from this study thus identified putative targets that are co-expressed with HER2 in colorectal cancer, which are different from HER2 in breast cancer, warranting further investigation into their role in colorectal cancer pathogenesis.

## Figures and Tables

**Figure 1 cancers-15-00130-f001:**
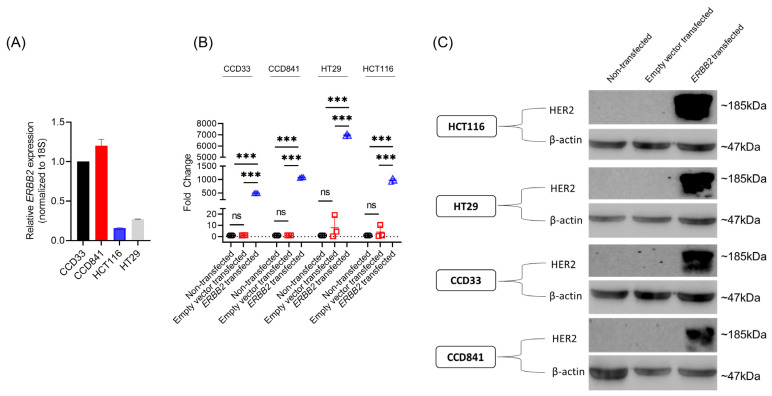
Validation of successful ectopic overexpression of *ERBB2* in the CRC (HCT116 and HT29) and normal colon (CCD33 and CCD841) cell lines. (**A**) Relative *ERBB2* expression in the normal colon cell lines CCD33 and CCD841 and the CRC cell lines HT29 and HCT116 as determined by qRT-PCR. Data were normalized to the expression of the internal control 18S rRNA gene and fold expressions were plotted relative to expression in the CCD33 cells. Data represent the mean ± SD of three independent experiments. (**B**) Relative *ERBB2* expression in non-transfected and either empty pcDNA3 vector or pcDNA3-*ERBB2* transfected HCT116, HT29, CCD33, and CCD841 cells as determined by qRT-PCR. Data were normalized to the expression of the internal control 18S rRNA gene and fold expressions were plotted relative to expression in the non-transfected cells. Data represent the mean ± SD of three independent experiments. *** *p* < 0.001; ns: not significant. (**C**) Same as B, but relative HER2 protein expression was determined in the different experimental conditions. Blots were re-probed with anti-β-Actin antibody to confirm equal loading across the lanes. The representative blots from three independent experiments are shown.

**Figure 2 cancers-15-00130-f002:**
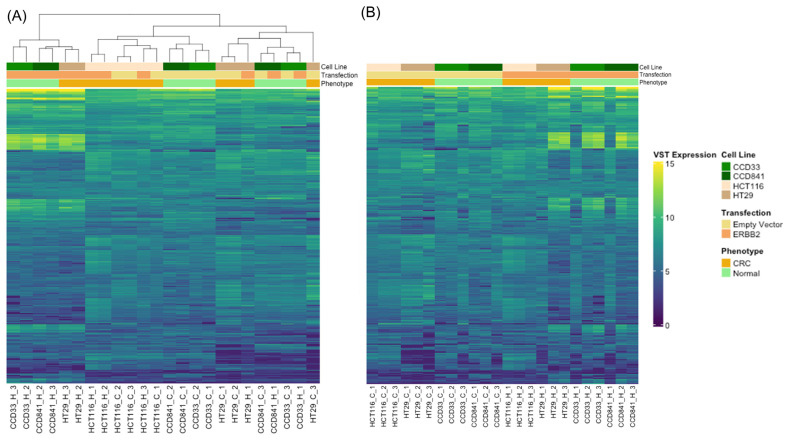
Heatmap of the differentially expressed genes in CRC (HCT116 and HT29) and normal colon (CCD33 and CCD841) cell lines transfected with empty vector or *ERBB2,* either clustered based on expression (**A**) or grouped based on transfection and phenotype (**B**).

**Figure 3 cancers-15-00130-f003:**
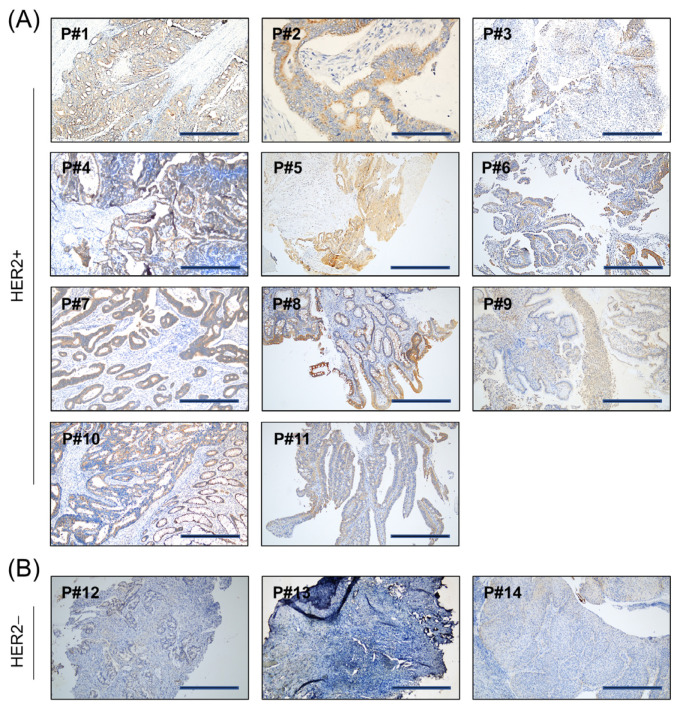
HER2 protein expression was detected by IHC staining in the 14 (P#1 to P#14) included patient samples. Images are representative IHC staining images of the 11 HER2+ (**A**) and 3 HER2− (**B**) patient samples. Scale bar, 100 µm.

**Figure 4 cancers-15-00130-f004:**
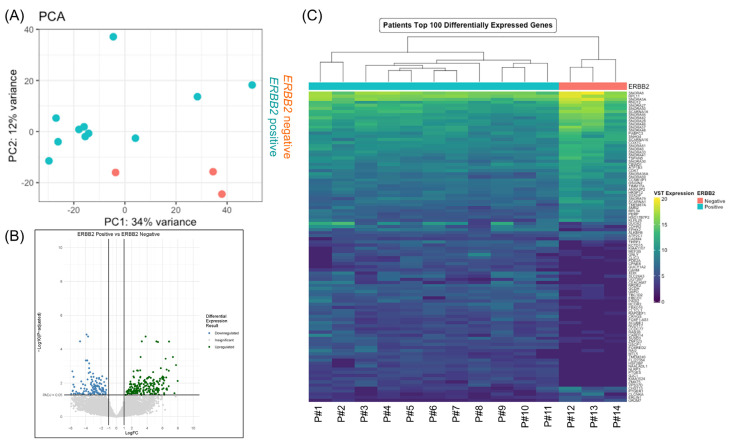
Genome-wide gene expression changes between *ERBB2*+ and *ERBB2*− CRC patients. (**A**) Principal component analysis (PCA) was performed to determine batch effects among the 14 patients’ samples. Comparison of PC1 and PC2 variation sequestered the samples based on *ERBB2* expression. (**B**) Volcano plot of differentially expressed genes between *ERBB2*- and *ERBB2*+ patients’ samples from input RNA-seq. Genes that are expressed significantly higher and lower based on log2 fold change in HER2+ samples are highlighted by green and blue dots, respectively. Unchanged transcripts are demarcated as grey circles (*p* > 0.05). (**C**) Heatmap of the top 100 differentially expressed genes.

**Figure 5 cancers-15-00130-f005:**
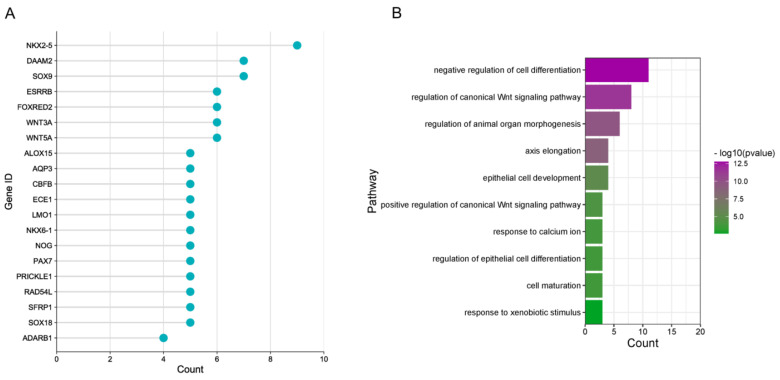
GSEA revealed significant enrichment of pathways related to cell differentiation and Wnt signaling. (**A**) The top 20 leading genes across the different enriched pathways based on frequency in HER2− vs. HER2+ CRC patients, and (**B**) the related pathways are shown.

**Figure 6 cancers-15-00130-f006:**
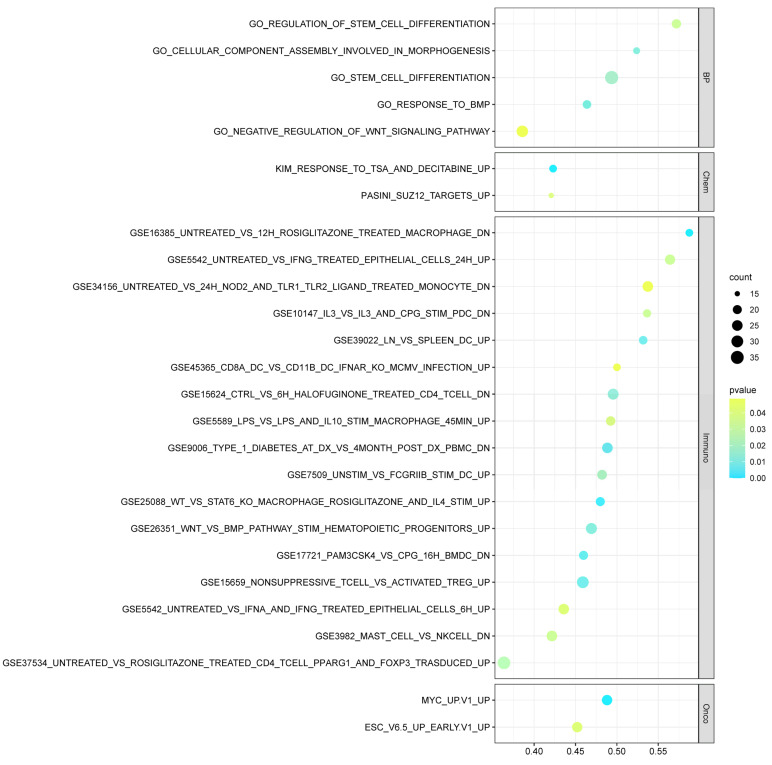
GSEA analysis of HER2+ and HER2− breast cancer and CRC patients revealed significant enrichment of common pathways. The figure shows the 26 significant pathways among CRC and BC in HER2+ vs. HER2− patients. *Immuno*, Immunologic Signature; *Onco*, Oncogenic Signature; *BP GO*, Biological Process; *Chem*, Chemical and Genetic Perturbations.

**Figure 7 cancers-15-00130-f007:**
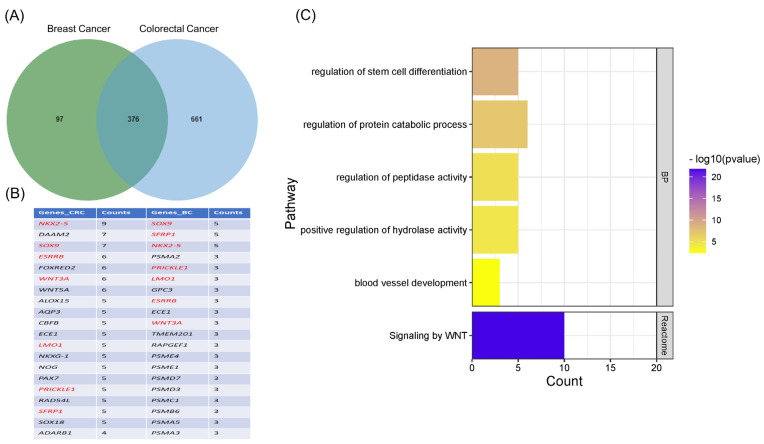
Top genes involved in the activated pathways of CRC and BC HER2+ vs. HER2− patients revealed a significant resemblance. (**A**) Venn diagram representing the overlap of genes overrepresented in enriched pathways in CRC and breast cancer patients. (**B**) Table showing the top 20 enriched genes in breast cancer and CRC patients. Genes shown in *red* were common in both CRC and breast cancer patients. (**C**) Related pathways of the top 20 enriched genes are shown. Representation is based on frequency in HER2− vs. HER2+ CRC patients.

## Data Availability

The RNA sequencing data generated in this work have been deposited in figshare and can be obtained from the following link: https://doi.org/10.6084/m9.figshare.21578763, accessed on 17 November 2022. The breast cancer datasets used in the manuscript are deposited in Gene Expression Omnibus (https://www.ncbi.nlm.nih.gov/geo, accessed on 3 October 2022) and are accessible through GEO Series accession numbers GSE29431 and GSE48391. All other supporting data of this study are either included in the manuscript or available on request from the corresponding author.

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
