# Peer review of "Transcriptomic Changes Associated with ERBB2 Overexpression in Colorectal Cancer Implicate a Potential Role of the Wnt Signaling Pathway in Tumorigenesis"

_cancers, 2022, doi:10.3390/cancers15010130_

Round 1

Reviewer 1 Report

In this study, Razzaq and colleagues aimed at detecting transcriptomic changes associated with ERBB2 expression in colorectal cancer (CRC). They analyzed different data, including a cohort of 14 patients with primary CRC, four cell lines and two additional breast cancer datasets. They found that overexpression of ERBB2 is linked with distinct transcriptional pathways, most importantly that of Wnt signaling and cell differentiation. 

Overall, the manuscript is well written, and the methodology seems sound, but the obvious weakness of the study is the small cohort size. Furthermore, several points in the manuscript require clarification, as there are parts where the methodology is not well described and rather confusing.

1) Be aware that all the figure legends are missing, making it difficult to evaluate the figures. Please fix this.

2) 243-254 and Sup. Fig 2A-D: For CCD33 and HT29, there seems not to be a distinct clustering of the HER2 overexpressing cells, please fix that statement.

3) Lines 246-251: It would be helpful to write how many genes were upregulated and how many downregulated. In the HT29, there seems to be very few genes upregulated in the HER2 overexpressing cells compared to the control and compared to the HCT116 cell line. Do you have any explanation why?

4) Figure 2. I have troubles seeing the “clearly subgroups of ERBB2 transfected and control cell lines” as you claim in the manuscript (line 250) from the actual expression values of the heatmap (and not from the color grouping on top of the heatmap). Maybe to make the figure more understandable, please scale the expression values so that it might show better the groups you mention. Also, please add the clustering tree of the samples.

5) Paragraph 3.2. Did you observe any difference in the clinical pathological data between the HER2+ and HER2- patients?

6) Figure 4A and 4C: Please keep consistent colours for the ERBB2 negative/positive in the PCA plot and the heatmap.

7) Figure 4B: All the non-significant genes are missing from the volcano plot.

8) Figure 4C: Maybe also scale the expression values to make the heatmap more readable. Also, by HER2 expression High/Low you mean ERBB2 positive/negative?

9) paragraph 3.3: The paragraph is a bit confusing to read and needs to be written more clearly. For example, first describe which pathways were enriched and then talk about the genes in those pathways. Also, maybe add in the supplementary material, the results of your GSEA analysis for each comparison, so that the reader can see for themselves the results (and those overlapping as well).

10) Line 319-320: Can the authors explain in more details how they split the patients in HER2+ and HER2? Why not using the HER2 IHC score to define if the patient was HER2+ and HER2-, as you have done for the CRC patients?

Author Response

We much appreciate the reviewer’s comments and thoughtful suggestions that have allowed us to improve our manuscript further. We attempted to carefully address all concerns of this reviewer and hope that the reviewer will find the revised manuscript suitable for publication in Cancers

Reviewer 2 Report

Transcriptomic changes associated with ERBB2 overexpression in colorectal cancer implicates Wnt signaling pathway and a potential role of NKX2-5 in tumorigenesis

In this manuscript, Razzaq et al. carry out a transcriptomic analysis of colorectal tumors presenting HER2 overexpression. Employing patient samples as well as cell lines with induced HER2 expression, the authors characterize cellular pathways and genes co-expressed with HER2. The authors provide a deeper understanding of the oncogenic role of HER2 in colorectal cancer and indicate differences with other cancers presenting HER2 overexpression such as malignancies of the breast.

The work described in this manuscript seems adequate for the scope of the journal and it is of interest for the cancer community. I would like to suggest a few changes that could strengthen this manuscript:

Fig. 1: How does short-term transfection (to an unphysiological expression level) compare to cell lines that have evolved under HER2-overepression selective pressure? Have the authors compared HER2-amplified or HER2-mutant cell lines to HER2-wt cell lines?

How long after transfection were the RNA samples isolated? The authors should indicate it in the methods section.

Fig. 4C: It seems like only half of the top 100 differentially expressed genes are different in the two groups and mainly are only overexpressed in the HER2 positive tumors. The first half of genes seems to be similarly overexpressed in both groups. Are these genes relevant?

Line 290: I would suggest the authors to briefly describe why they chose these collections.

Line 379: Should it be NKX2-5 instead?

Line 384 and 387: It gets very confusing if the authors keep changing the name of this transcription factor.

Comment: APC loss-of-function mutations are frequently observed in CRC and result in hyperactivation of the Wnt signaling pathway. Have the authors analyzed whether APC mutations and HER2 amplification/overpression co-occur? Are they mutually exclusive? I am wondering whether HER2 overexpression could result in a similar phenotype to APC loss (I don’t think the authors have mentioned APC at all in this manuscript).

Comment: The authors should show direct effect of NKX2-5 on Wnt signaling pathway activation in the context of normal or high HER2 expressing colorectal cancer cells (knockdown, overexpression experimental assays to say the least). Particularly if they want to claim it on the manuscript title. The authors did not show any data supporting “a potential role of NKX2-5 in tumorigenesis”.

Author Response

We are very thankful for the reviewer’s comments and thoughtful suggestions to improve the manuscript. We attempted to address all the concerns raised and updated the manuscript accordingly.

Round 2

Reviewer 1 Report

The authors have satisfactory replied to the questions and suggestions of the reviewers and have modified their manuscript accordingly. I have no further comments. Congratulations for your work.

Reviewer 2 Report

The authors have sufficiently modified the manuscript to incorporate suggested changes by the reviewers. Their work is now better suited for publication. 
Thank you for your hard work.